# Probabilistic edge weights fine-tune Boolean network dynamics

**Dávid Deritei**[1,2]*, **Nina Kunšič**[1], **Péter Csermely**[1]

**1** Department of Molecular Biology, Institute of Biochemistry and Molecular Biology, Semmelweis University, Budapest, Hungary, **2** Channing Division of Network Medicine, Brigham and Women's Hospital, Harvard Medical School, Boston, United States of America

\* david.deritei@channing.harvard.edu

## Abstract

Biological systems are noisy by nature. This aspect is reflected in our experimental measurements and should be reflected in the models we build to better understand these systems. Noise can be especially consequential when trying to interpret specific regulatory interactions, i.e. regulatory network edges. In this paper, we propose a method to explicitly encode edge-noise in Boolean dynamical systems by probabilistic edge-weight (PEW) operators. PEW operators have two important features: first, they introduce a form of edge-weight into Boolean models through the noise, second, the noise is dependent on the dynamical state of the system, which enables more biologically meaningful modeling choices. Moreover, we offer a simple-to-use implementation in the already well-established BooleanNet framework. In two application cases, we show how the introduction of just a few PEW operators in Boolean models can fine-tune the emergent dynamics and increase the accuracy of qualitative predictions. This includes fine-tuning interactions which cause non-biological behaviors when switching between asynchronous and synchronous update schemes in dynamical simulations. Moreover, PEW operators also open the way to encode more exotic cellular dynamics, such as cellular learning, and to implementing edge-weights for regulatory networks inferred from omics data.

## Author summary

The life and decision-making of cells is regulated by a complex web of dynamically interacting molecules. The strength and nature of individual interactions is very diverse, and it is especially important to understand such diversity when it comes to defects and disease. For example, the mutation of a protein binding site can critically alter the probability and strength of its interactions with its binding partners. Boolean network models have become an increasingly potent tool for understanding the complex dynamical interactions within cellular regulatory systems, however, there is no straightforward and explicit way to encode weights on individual interactions. In this paper we offer a way to add weights to interactions by simple noise operators which alter the behavior of edges (or groups of edges) in in-silico simulations of Boolean network models. We show with multiple applications that adding just a few PEW (probabilistic edge-weight) operators dramatically

**Data Availability Statement:** The modified version BooleanNet along with the Jupyter notebooks containing the code for all the results presented in this paper, as well as the ways to reproduce some of the other noisy models are available on the

following GitHub page: https://github.com/deriteidavid/boolean2pew. (S1 Notebook: Murrugarra_et_al_2012_paper_results.ipynb, S2 Notebook: Poret_et_al_paper_results-Boolean_init.ipynb, S3 Notebook: Supplementary_Application_Note.ipynb, S4 Notebook: Applications.ipynb).

**Funding:** This work was supported by the Hungarian National Research, Development and Innovation Office (K131458), by the Higher Education Institutional Excellence Programme of the Ministry of Human Capacities in Hungary, within the framework of the Molecular Biology thematic programs of Semmelweis University, by the Thematic Excellence Programme (Tématerületi Kiválósági Program, 2020-4.1.1.-TKP2020, TKP2021-EGA-24) of the Ministry for Innovation and Technology in Hungary, within the framework of the Molecular Biology thematic program of the Semmelweis University. All authors, DD, NK, and PCs received salaries from the funding while developing this work. The funders had no role in study design, data collection and analysis, decision to publish, or preparation of the manuscript.

**Competing interests:** The authors have declared that no competing interests exist.

improves the biological plausibility of Boolean models and reproduces much more nuanced experimental results.

This is a *PLOS Computational Biology* Methods paper.

## Introduction

Boolean network models or Boolean dynamical systems have become a standard toolkit for modeling biological systems of increasing size and complexity [1–7]. The main advantage of using Boolean models is that they offer a reasonable compromise in complexity: the regulatory mechanisms and interactions are expressed through logical rules, while the number of parameters remains manageable, even in large systems. The dynamical side of Boolean models is key: the attractors of validated models correspond to stable phenotypes of biological systems [8,9]. Such a representation of phenotypes can help to understand the underlying mechanisms of the behavior such as biological phenotypes emerging from local interactions, mutations leading to pathological phenotypes, etc. [10,11]. Boolean models can also help in identifying the key regulatory circuits associated with phenotypic decision-making [12,13] and even help identify control targets in order to drive the system into phenotypes (e.g. to switch from an unhealthy state to a healthy one) [14–16]. Boolean models address questions that lie at the core of modern systems biology, but also of modern medicine and drug development [2].

A large family of Boolean Network models is concerned with modeling biological noise and uncertainty coming from incomplete measurements or sparsity of data. Some of these methods introduce uniform noise on a system level (e.g. perturbed Boolean Networks), introduce noisy function selection on the level of nodes (e.g. Probabilistic Boolean Networks [17], Dynamic Bayesian Networks [18], Stochastic Discrete Dynamical Systems [19]), or combine system- and node-level noise (perturbed Probabilistic Boolean Networks) [20]. In this work, we introduce a method that applies noise on the level of individual edges or hyper-edges (edges defined between sets of vertices) in a biologically meaningful way, through probabilistic edge weight (PEW) operators. Generally, PEW operators are mathematical objects that can be added to the regulatory rules of Boolean models to modulate the noisiness of edges. On one hand, PEW operators offer a way to handle edge-level uncertainty, on the other, they offer a means to introduce relative edge-strength within a system. Moreover, the noise level can be made a function of the system's dynamical state, which is not a common way of introducing noise in Boolean systems, despite the evidence in literature that in fact noise can be highly dependent on the dynamical state and environment of the cell [21–23]. For example the dynamical behavior of a node can fluctuate when its regulators are present but their molecular interactions are noisy. In contrast, it's less likely to fluctuate from an off state when its regulators are absent. In the next sections, we place the PEW operator framework within the context of other noisy Boolean models and demonstrate through two empirical examples that introducing PEW operators to the Boolean model improves its predictive capabilities.

## Methods

### Boolean dynamical systems—Definitions

Boolean regulatory networks can be represented by a graph $G = (V, E)$ consisting of $V = (v_1, v_2, \ldots, v_N)$ vertices and $E = (e_{ij} | i, j \in V)$ directed edges. Each vertex (node) has a binary state, $\sigma_v$

equal to 1 or 0, often referred to as ON or OFF. The state of the system is the collective state configuration of all of its constituent nodes in time $t$. Overall, the model can have $2^N$ different states, where $N$ is the number of vertices (nodes). The state of each node $v$ is determined by a unique logical function assigned to it, $F = (f_1, f_2, \ldots, f_N)$. We also call these functions Boolean regulatory functions or Boolean rules. The logical function encodes how every node responds to the different state-combinations of its regulators. The inputs of each function $f_i$ are the states of the regulators (nodes with edges pointing to $v_i$) of $v_i$ in time point $t$. The value of a node $\sigma_{v_i}$ in time-step $t + 1$ is calculated as:

$$\sigma_{v_i}(t + 1) = f_i\left(\boldsymbol{\sigma}_{Par(v_i)}(t)\right),$$

where $Par(v_i)$ is the set of parents (regulators) of node $v_i$ and $\boldsymbol{\sigma}_{Par(v_i)}(t)$ is their state configuration at time-point $t$.

Different *update schemes* determine the order in which the functions are evaluated, i.e. the state of the nodes is updated. In the case of the *synchronous* update scheme, all nodes are updated at the same time, such that the state of the system in time $t$ fully determines the state of the system in time $t + 1$. This results in deterministic trajectories, where the emergent dynamics of the system depend only on the initial configuration. *Asynchronous* update schemes update the system one node at a time and emulate different (more granular) time scales as compared to synchronous update schemes. This is mainly because in the case of synchronous update one time-step represents $N$ node updates, while in the case of asynchronous updates, one time-step represents only one node update. There are exceptions when $t$ is incremented only after all nodes were updated at least once asynchronously. A certain degree of stochasticity can be added to the system dynamics with randomized update schemes. *Random order asynchronous* update picks nodes from a shuffled list of the nodes, updating each once before reshuffling, while *general asynchronous* update picks nodes in a random way, allowing repetition.

The *attractors* of a Boolean system represent the long-term equilibrium states of their dynamics. Fixed-point attractors or *steady states* are states in which all logical functions are satisfied and updating nodes no longer changes the state of the system. Attractors can also be limit cycles or complex attractors, which, instead of a single state, are a set of states that the system keeps visiting indefinitely (a.k.a. an ergodic subset of states).

## Boolean models with stochastic properties introduce different varieties of noise

Scientists have realized early on that completely deterministic Boolean networks (BNs) are limited in their capability to model real systems [2,17,19,20,24]. There is indeed a spectrum of noise sources in a biological system, from basic thermodynamic noise to heritable genetic differences (mutations) among individual cells in a modeled cell population; there is also measurement noise as well as uncertainty due to lack of data [25,26]. Mirroring this variety in sources of noise in real systems, there are many types of noise one can introduce into a BN.

One of the earliest types of noisy BN were the perturbed Boolean Networks (BNp) first suggested in [27] as Kauffman networks with "thermodynamic noise". In this paper we use the technical description used in the review article by Trairatphisan et al. [28]. The noise in BNps is added by a perturbation of a random node-flip with a nonzero probability $p$ in each time step. Essentially, before each node update one tosses a coin with bias $p$, which determines whether the next value of the node is going to be determined by its deterministic function or

it's going to flip to its other state, regardless of its function. This simple modification turns BNs into ergodic Markov Chains, which adds a number of advantages to their use and analysis. For instance, attractors in BNps still carry most of the probability mass but the system can leave the attractors with a nonzero probability. One of the drawbacks of BNps is that they introduce a very general "thermodynamic" stochasticity that does not give much freedom in fine-tuning node-specific aspects of the noise.

One of the most prominent families of noisy BNs are Probabilistic Boolean Networks (PBNs) introduced by Shmulevich et al. [17], which address the lack of node-specificity of BNps. PBNs have a set of functions assigned to each node (instead of a single unique logical function). The different possible combinations of selected functions give rise to different "real-izations" of BNs. The total number of realizations is equal to the product of function set sizes. At each time update, a random binary variable $\xi$ decides whether a new realization shall be used, or the BN remains the same. If $\xi(t) = 1$ new functions are chosen for each node from their individual pool of functions in each time-step. The functions from each set are picked with a pre-determined probability (adding up to 1 within each set).

PBNs do not necessarily constitute ergodic systems, however, they can contain ergodic sub-sets of states akin to complex attractors. One can also combine perturbation noise with PBNs (PBNp), this way there is a non-zero probability of nodes committing "mistakes" and also changing their function. PBNs have a wide range of applications and are a popular model of biological systems [28].

PEW models and PBNs can be mathematically equivalent in certain conditions. We discuss the relationship between the PEW and the PBN frameworks in the S1 Document.

Another popular noisy model that incorporates a dependence of the noise on the system state is the stochastic discrete dynamical system (SDDS) introduced by Murrugarra et al. [19]. Instead of a set of functions (like in PBNs), nodes in SDDSs are assigned a single function and two probability values: an activation and a degradation propensity ($p_{up}$ and $p_{down}$). The two propensity values determine whether a node "accepts" its new update value or remains the same. If a node's current value $x(t)$ is smaller than its update value, $f(x(t))$, meaning $x$ is up-reg-ulated, then the up propensity ($p_{up}$) will determine by a biased coin-toss if the $x(t + 1)$ will be equal to $f(x(t))$ or to $x(t)$. The case of down-regulation works the same way when $x(t) > f(x(t))$, with $p_{down}$ determining the bias of the coin-toss.

A significant innovation of SDDSs, as alluded to earlier, is that the noise is dependent on the value of the node at time $t$, and also the probabilities are not necessarily symmetric (e.g., $p_{up}^i + p_{down}^i \neq 1$). This might not carry the same mathematical elegance as previous models but is very useful biologically, where such asymmetries are common. The method presented in this paper has a similar principle as the one introduced by Murrugarra et al. [19] in that the noise is dependent on the state of the system and is not necessarily symmetrical.

Murphy et al. have shown [18] that Boolean networks are indeed a special case of a broader class of Dynamic Bayesian Networks (DBNs). DBNs are defined by a set of nodes that repre-sent random (hidden or known) variables, and directed links are described by the conditional dependence between the variables. The value of each node is determined by a conditional probability distribution (CPD), dependent on the parents of the node. Dynamic Bayesian net-works are a special case of the general Bayesian networks, where the dynamic aspect is encoded by a different set of $N$ nodes representing each timestep. If all nodes have deterministic logical functions with a binary output, then the DBN is a Boolean network.

We would argue that the PEW method is a step toward the DBN framework in generality because the rules of some nodes with PEWs become stochastic and conditionally dependent on the value of the (subset of) parents. Nonetheless, it's still more pragmatic to view PEW

models as a separate framework because, as we show in the empirical applications, probabilistic edge weights are meant as very specific, targeted modifications to Boolean models and are easily made compatible with existing tools of Boolean network analysis. On the other hand, DBNs are likely the most general modeling framework of which PEWs models represent a special case.

Finally, the work of Poret et al. [29] is worth mentioning, even though their paper has not been published in a peer-reviewed journal. Their method is close in spirit to what is proposed in this paper, and they provide great examples of fine-tuning BN dynamics with different kinds of noise. In their work Poret et al. introduce "fuzzy operators" in their logical functions, which can evaluate continuous node-states. The use of "fuzzy operators" allows the method to fine-tune edge responsiveness and edge reactivity, all of which are indeed quite nuanced interactions.

We have used a similar weakening/strengthening approach on the level of nodes in one of our recent papers [30] and its follow-up study [31], where we imposed external perturbations to specific nodes, especially input nodes, setting them to a certain Boolean value with probability $p$ in every time-step. This helps to establish a continuous "concentration"-like parameter, which can influence the downstream dynamics to a great degree. We use this same approach in this paper to alter the input concentration of TGFB in the applications to the EMT model.

In the following sections, we present the PEW method and we show a few empirical examples in which we demonstrate its usefulness in recapitulating and explaining experimental results. Finally, we demonstrate its versatility through reproducing examples from other stochastic methods. Indeed, BNp-s, SDDS (Boolean case), and the simulation results of Poret et al. [29] are all reproducible in the PEW framework (See S2 Document, S1 and S2 Notebooks).

## The PEWs offer an easy way to insert noise to individual edges in a Boolean network

Let us denote the probabilistic weight operator as $P_e$, which has two parameters: $w_{on}$ and $w_{off}$. We also associate a function $f$ to $P_e$, which describes how the weights determine the outcome of the operation.

Generally:

$$P_e\Big(f, \, w_{on}, \, w_{off}\Big)x = \begin{cases} f(x, \, w_{on}), & if \ x \, > \, \theta \\ f\Big(x, \, w_{off}\Big), & if \ x \, \leq \, \theta \end{cases} \tag{1}$$

Here $x$ represents the source of a (hyper)edge in the Boolean model. A hyperedge is an edge of a hypergraph, a generalized graph, where edges connect sets of vertices (undirected) or ordered subset pairs (directed) [32]. In a Boolean model, a hyper-edge is a clause of the Boolean function consisting of multiple nodes regulating the target node (e.g. A AND B). The $P_e$ operator used in a Boolean rule affects the first variable on its right-hand side. This variable can refer to a single node or to a clause of the Boolean rule involving multiple input nodes (in which case the operator is followed by parentheses, containing a subset of the logic function). Such clauses represent hyperedges, as they characterize the relationship between the target node and several of its inputs. The $\theta$ threshold parameter is 0.5 for the Boolean case, but for continuous or multi-level cases it can be adjusted.

The "grammar" of the $P_e$ operator is the same as of any left-hand side operator (such as "not") i.e. it acts on the first mathematical clause to its right. Below we illustrate the rationale and the exact details of how the PEW operator works in a simple example.

Consider the Boolean rule A * = B AND C. In biological models, a function such as this usually encodes that gene A is activated by a protein-complex formed by B and C. Both B and C are *necessary* for A's activation; thus, A will not be activated if C = 0. Next, we assume that the link between A and C is weakened (e.g., due to a mutation), in other words, we decrease the weight of the C → A link. Adding a PEW operator we can represent this as $A^* = B$ AND $P_e$ C. The weight is decreased in a probabilistic way by making it noisier. The most extreme case of weakening an edge is cutting it entirely. In general, cutting a link in a Boolean network implies a choice between setting the state of the source of the link to 0 or 1. These two choices have drastically different interpretations. Cutting the C → A link by setting C = 1 yields the Boolean rule $A^* = B$, which indicates that the necessity of C was eliminated altogether. If the C → A link is cut by setting C = 0 then the Boolean rule becomes $A^* = 0$, thus A will be locked OFF. The mechanistic interpretation of $A^* = B$ AND C helps explain the *need to apply different levels of noise to the ON vs. OFF states of C*. For instance, if C is mutated so that it only binds B 30% of the time, this probability should only be applied to the cases in which *C is ON*, i.e. there is something to bind to B. If C is OFF there is nothing even to attempt the complex formation, therefore A will be OFF *100% of the time*. This is the reasoning behind the conditional update in Eq (1). In this example $w_{on}$ = 0.3, and $w_{off}$ = 0.

As alluded to in the example above *f* can be a simple draw from a binary distribution (i.e. a biased coin-toss) with probability *w*,

$$f(x, w) = Pr\{x = 1, w\}.$$

In summary, the new rule is $A^* = B$ and $P_e(f, w_{on}, w_{off})$ C. This means that whenever C is ON the noise weakens its effect on A as though it were ON only 30% of the time and whenever it is OFF it is not affected by noise.

Finally, the PEW operator is not restricted to acting on single edges, but it can act on any hyperedge. This means practically, that any clause of the Boolean rule can be made noisy with the PEW operator, and any number of PEW operators can be used. In our previous example, it is possible that the B+C complex itself is noisy. In that case, $A^* = P_e(B$ and C) would make the joint effect of B and C noisy instead of a single node's.

The exact mathematical framework of the PEW approach depends on the noise function applied in the operators, nonetheless, adding any kind of noise to a single edge can make the target node stochastic. Stochastic nodes indeed transform the state transition graph (STG) of the Boolean model into a discrete Markov chain. However, just as in the case of Probabilistic Boolean Networks, the Markov chain is not necessarily ergodic, but it can have ergodic subsets, due to the fact that only a subset of nodes is stochastic and other non-stochastic parts of the network can still lock into states permanently. For the purposes of our paper, we chose the representation/analysis of the dynamic evolution of models as ensemble averages of states (started from specific, biologically relevant initial conditions), instead of a more general STG/Markov chain analysis (e.g. determining the stationary probability of states within ergodic subsets, PageRank, etc.). We do this mainly because the ensemble averages capture relevant time evolution patterns and the stochasticity in the case of certain nodes emerges as a nonbinary average in the steady state, akin to intermediate concentrations of molecules in cells.

The application of targeted PEW operators on an ensemble of simulated systems leads to the fine-tuning of the concentrations of molecules, which otherwise behave non-biologically in the model. In the next sections, we will show the biological relevance of such fine-tuning.

### Implementation in BooleanNet

One of our goals with this method is to offer an easy way to use PEW operators without the need for additional software. Thus, the technical implementation is done as an extension of the already popular and well-established BooleanNet framework in Python [33]. In the modified version of BooleanNet PEW operators are implemented within the rule-parsing grammar of the BooleanNet method (which uses the Yacc framework [34]) and work as left-hand side operators (the same as **not**). The syntax of a PEW operator in a Boolean rule is two positive floating-point numbers [$w_{on}$, $w_{off}$], separated by a comma within a square bracket. Following the example used previously:

A* = B and C after applying the PEW operator becomes:

A* = B and **[0.3,0]** C

Or using the hyperedge:

A* = **[0.3,0]** (B and C)

To use a PEW operator in BooleanNet only the Boolean rules of the simulated model have to be modified, following the syntax specified above. All other functions and operations (unless defined otherwise) follow the BooleanNet standards. The PEW operators also interact intuitively with the **not** operator, so using a PEW operator before or after a **not** has different effects. For example:

A* = **[0.3, 0]** not C

has the same probabilistic outcome as

A* = not **[0.7,1]** C.

Generally, the **not** operator to the left modifies a PEW operator [$w_{on}$, $w_{off}$] to [$1 - w_{on}$, $1 - w_{off}$]. A PEW operator to the left of a **not**, on the other hand, will act on the outcome of the negation.

In this paper we follow the logic of the first number being associated with the ON case of the clause acted upon ($x$) and the second number is associated with the OFF case. This, however, can be easily changed in the implementation, along with the threshold parameter $\theta$. Moreover, future versions of the framework can be expanded to the multilevel case, where instead of two values one can have a vector of arbitrary length, corresponding to all possible values of the input nodes.

Using the [$w_{on}$, $w_{off}$] syntax, the code defaults to the Bernoulli coin-toss as the noise function. Yet generally one can also specify the noise function in the operator: [**f_name, $p_1$, $p_2$**], where the "f_name" is a function defined in a separate Python file and $p_1$, $p_2$ are parameters of the function. The parser interprets each operator as a separate entity, so operators with different noise functions can be mixed in the same model. To see use cases of this general formula please see the Application Note (S3 Document) and its accompanying Jupyter notebook (S3 Notebook).

The parameters and the noise function of the PEW operators are *not* time dependent. Nonetheless, it is technically possible to make them time-dependent (by changing the parsed rules during a simulation) but we don't do this in any of our applications. A PEW operator applied in the rule of a single node will potentially make the *target node* stochastic. This also means that all nodes downstream of the stochastic node can behave stochastically as a result. The combination of noise operators with the complex nature of regulatory networks can have non-trivial emergent effects, as we see in the applications.

## Results

In this section, we are going to demonstrate some applications of the PEW model, where the PEW enhanced BN performs better in explaining experimental results than the classic Boolean

model, yet no complex ODE model is needed. To do this we switch from the more general $P_e(f, w_{on}, w_{off})$ notation to the more pragmatic $[p_{on}, p_{off}]$ notation, where the noise function is the Bernoulli coin-toss and the weights are determined as probabilities.

## Noisy feedback-loops explain the loss of M attractor stability in epithelial to mesenchymal transition (EMT)

The transitions between epithelial and mesenchymal cellular phenotypes are encountered in embryonic development, wound healing, and cancer metastasis [35–37]. One of several proteins that trigger the signaling network that leads to epithelial-mesenchymal transition (EMT) is Transforming growth factor-beta (TGFB), an extracellular signaling molecule, which can trigger EMT but is also secreted by cells that underwent EMT. At the core of the EMT is a mutually repressive positive feedback loop between the Zeb transcription factors expressed in mesenchymal cells (M state), and the miR-200 microRNA expressed in epithelial cells (E state). This system acts as a bistable switch; once flipped from E to M, cells lose expression of E-cadherin, a critical adherens junction molecule required for forming epithelial monolayers and maintaining an epithelial phenotype [6,16,38]. Celia-Terrassa et al. [39] showed in a fascinating study that a single mutation that weakened Zeb's ability to repress miR-200 expression radically altered the commitment dynamics of TGFB-induced EMT, and sped up the mesenchymal to epithelial transition (MET). Here we propose a very simple Boolean model, which qualitatively reproduces most of the results of the Celia-Terrassa study. Moreover, we show that the addition of two PEW-operators to the model significantly improves the model's qualitative results.

First, Celia-Terrassa et al. [39] showed that with increased TGFB concentration their cell lines exhibited a bistable behavior where the epithelial marker E-cadherin showed two distinct concentration peaks; one at a high concentration associated with a population of epithelial cells and one at a low concentration of a population of mesenchymal cells. This can be explained by the lock-in of the Zeb—miR200 mutual inhibition loop, which becomes self-sustaining (hysteresis). In Boolean modeling, we identify self-sustaining positive feedback loops as *stable motifs* [12], patterns of node activation that permanently lock in within the dynamics of the system. The authors created a separate cell line, where, using CRISPR technology, they weakened the Zeb-miR200 feedback loop by mutating the binding site of Zeb1 on miR200. This led to the disappearance of the previously observed bistability, where the E-cadherin peak changed linearly with increased TGFB concentration (instead of switching from low to high and never stabilizing between the two bistable peaks). In the following text, this Zeb1-miR200 edge knock-out line variant will be simply referred to as "mutant".

Second, they showed that in wild-type cell lines even a short (5 min), high concentration TGFB pulse can commit cells to EMT (even after washing off TGFB). The commitment only happens in mutant cells after a much longer (1 h) pulse, suggesting some other self-sustaining feedback loop downstream (other than the ZEB-miR200), which gets activated only with a longer pulse. Third, the authors showed that mutant cells left unperturbed transition back into the epithelial state much quicker (~3–6 days) than wild-type cells (~12 days). The authors proposed a simple 4 node ODE model which explains the bistable behavior due to the ZEB-miR200 feedback loop and other related phenomena [39]. In this paper, we show that most results can also be explained using a simple PEW-enhanced Boolean model, with a minimal number of parameters.

There are a number of well-established Boolean models for EMT [6], however, for the reproduction of these results we propose a simplified model with an important update to previous models, namely, that TGFB has to have a shorter, direct path to ZEB, which does *not*

involve SNAI1 (in Steinway et. al. [6] all TGFB—ZEB paths go through SNAI1). The evidence that supports this shortcut is clear in the order of activation in response to the TGFB signal reported in the Celia-Terrassa paper (Fig 3C in [39]): SNAI1 turns on 5 hours after ZEB1. In our model, we represent this delayed signal from TGFB to SNAI1 with $n$ artificial intermediary nodes. We model the delayed feedback loop suggested by the experimental evidence with a SNAI1 local positive feedback loop; such feedback loop has support in the literature (see S4 Document). We also add an alternate inhibitory pathway from TGFB to miR200 with $m < n$ intermediary nodes. For the Boolean rules of the model and additional references see S4 Document.

One potential disadvantage of Boolean models is that self-sustaining feedback loops (stable motifs) lock in permanently in the model dynamics, not encapsulating the fact that often there is a natural decay in the self-sustaining nature of the feedback loop after the termination of the initial signal. Simulating the model proposed above with classical Boolean dynamics permanently locks it into the mesenchymal state, given a long enough initial TGFB pulse. With the PEW framework, we can make stable motifs slightly more reversible, resulting in more nuanced dynamics. In the "wild type" version of our model, we add a very slight noise to the edges highlighted in red in Fig 1, namely to the local feedback loop of SNAI1 and the ZEB—miR200 link. The consequence of this noise is that after the termination of the TGFB signal the system will slowly converge back into the epithelial state (i.e. it will undergo MET). Simulations with an ensemble of Boolean networks are shown in Fig 2. In Fig 3 we show that despite the edge noise the bistability due to the positive feedback loop is still maintained, i.e., the noise does not destroy the nonlinear effect of the transition.

In the "mutant" version of our model instead of severing the ZEB—miR200 edge completely we create an *almost* severed noisy edge, assuming that the mutation does not fully

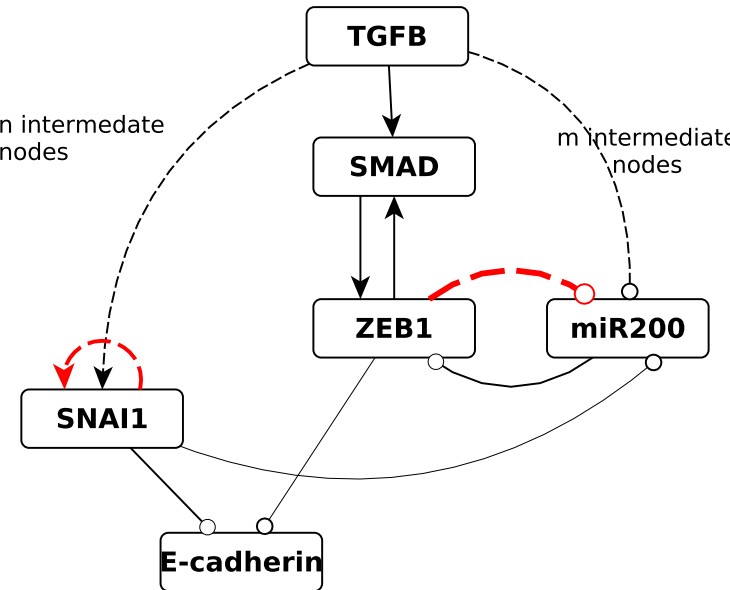

**Fig 1. A simplified Boolean model of the EMT with PEW edges.** Compared to another popular Boolean model of the EMT [6] this version has a shortcut from TGFB to ZEB (independent of SNAI1) which we implemented through SMAD (evidence for a direct path in [40]). This shortcut is also justified by SNAI1's delayed activation compared to ZEB after the initial TGFB pulse in [39]. Edges ending in circles represent inhibition, edges ending in arrows represent activation. The black edges with dashed lines represent chains of intermediate dummy nodes which emulate the delayed signal (n = 6, m = 3). The red dashed edges represent the feedback edges that have noise operators applied to them in both the wild type and the mutant case.

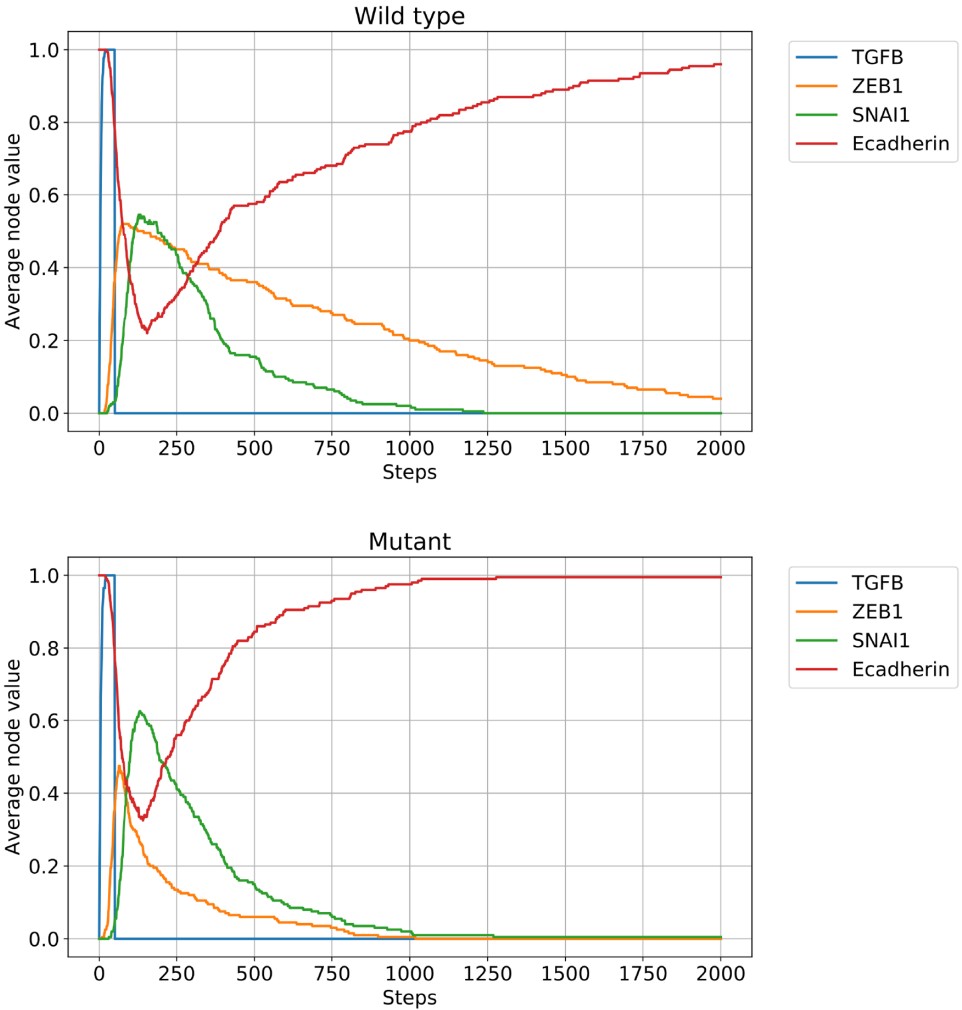

**Fig 2. The "mutant" model has a faster MET compared to the wild-type version.** Both panels represent the average node values of 200 independently simulated EMT models with a single strong initial TGFB pulse. The wild-type model (top) has a slight noise on both edges highlighted in red in Fig 1 ($p_{on}$ = 0.95, $p_{off}$ = 0). The mutant (bottom) version has a stronger noise on the Zeb—miR200 edge ($p_{on}$ = 0.05, $p_{off}$ = 0) making the link almost severed. Due to the noise on the feedback loops, both assemblies return to the epithelial attractor, but the mutant does it a few hundred steps sooner due to the faster loss of ZEB, qualitatively matching the experimental results of [39].

abolish Zeb's ability to repress miR200. This is also a great example of why the noise should be dependent on the state of the node: When ZEB1 is ON there is still a slight chance for it to inhibit miR200 (imperfect loss of repression due to the mutation), but when it is OFF (e.g., in the absence of a TGFB signal), the chance of it turning miR200 OFF should be 0 (as is in the wild-type case).

Figs 2 and 3 show two additional model results that recapitulate the experimental data: first, the MET is significantly quicker in the mutant system compared to the wild type, and second, the mutant version loses its bistability, shown by the red median line of the E-cadherin distribution averaged in the interval of timesteps between 100 and 150. However, the small amount of added noise in the "wild type" version on the edge ZEB—miR200 edge preserves the bistable transition with the increased TFGB concentration (Fig 3 left panel) we expect with

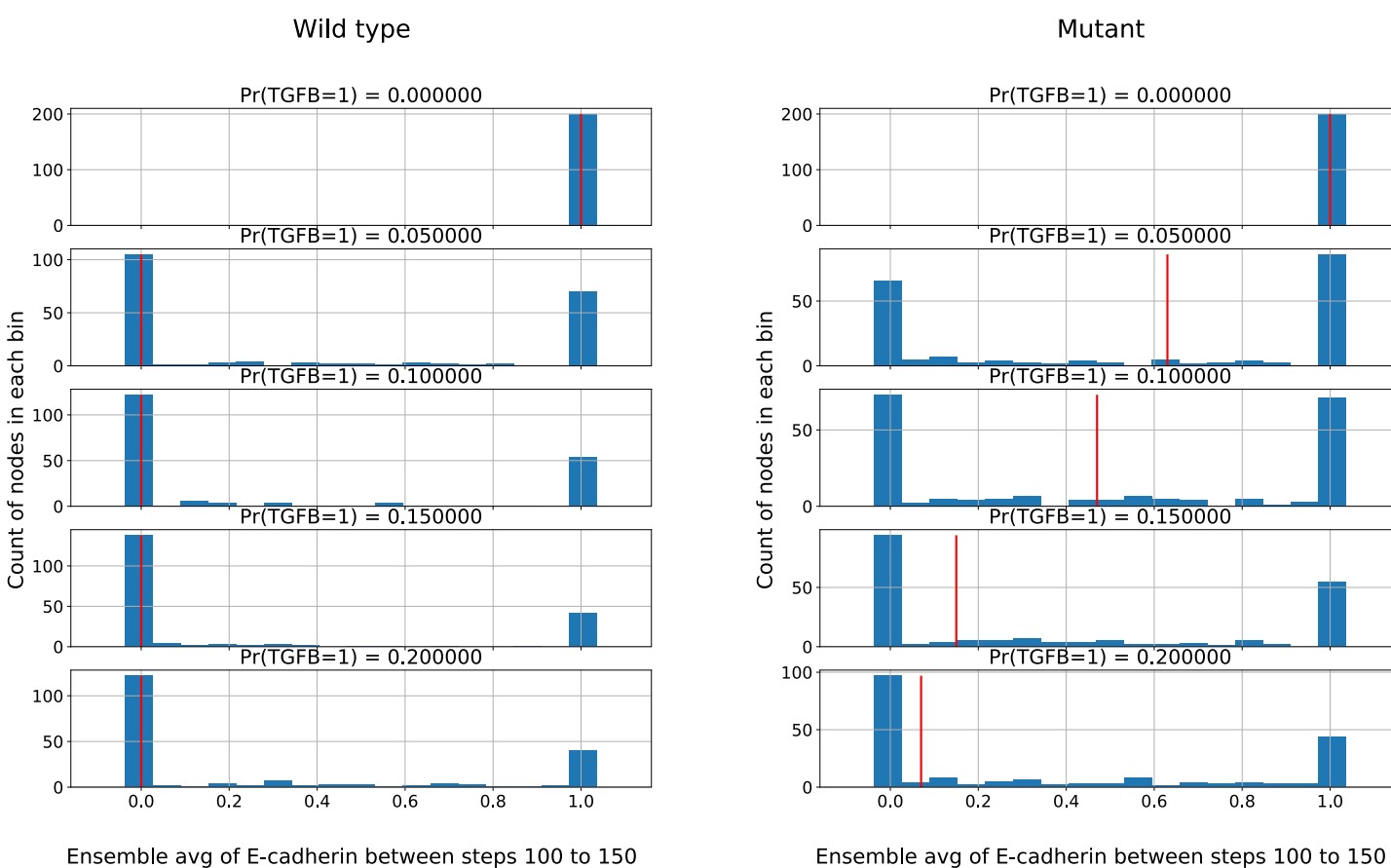

**Fig 3. The mutant EMT model loses its nonlinear bistability in the function of the TGFB pulse concentration.** The figures show the distribution of average E-cadherin between steps 100–150 in the models of wild type (left) and mutant cells (right). The red vertical line represents the median of the distribution. Despite the noisy edge the wild type still exhibits a sudden bistable transition between epithelial and mesenchymal states. The mutant, however, shows a much more gradual transition. This also fits some of the experimental results of [39]. This result also shows that self-sustaining feedback loops (stable motifs) don't necessarily lose their nonlinear attributes with some added noise.

a noiseless stable motif, while still allowing the MET to happen because the feedback loop is not irreversibly locked in. The initial state of the simulated networks is the epithelial attractor (Ecaderin = miR200 = ON; all other nodes OFF).

## Reduced CyclinB-Cdk1 induced apoptosis improves asynchronous cell cycle model

In 2019 we published an 89 node modular Boolean cell cycle model, which explained a wide range of healthy and pathological cell cycle behaviors, from the aberrant cell cycle driven by hyperactive PI3K, to the different effects of timed knockout of Polo-kinase 1 (Plk1), such as mitotic catastrophe or polyploidy [30]. The model has the most robust cyclic behavior when simulated with the synchronous update scheme. In Sizek et al. [30] we also considered multiple kinds of asynchronous update schemes and found that the model had a few non-biological behaviors when simulated with the general asynchronous update scheme. These behaviors were resolved when using a biased order asynchronous update scheme, which guaranteed that a critical subset of node updates are made in their biologically observed order. However, the biased order asynchronous update scheme needs a number of additional parameters and its implementation is rather non-intuitive and hard to link to biology in a straightforward way.

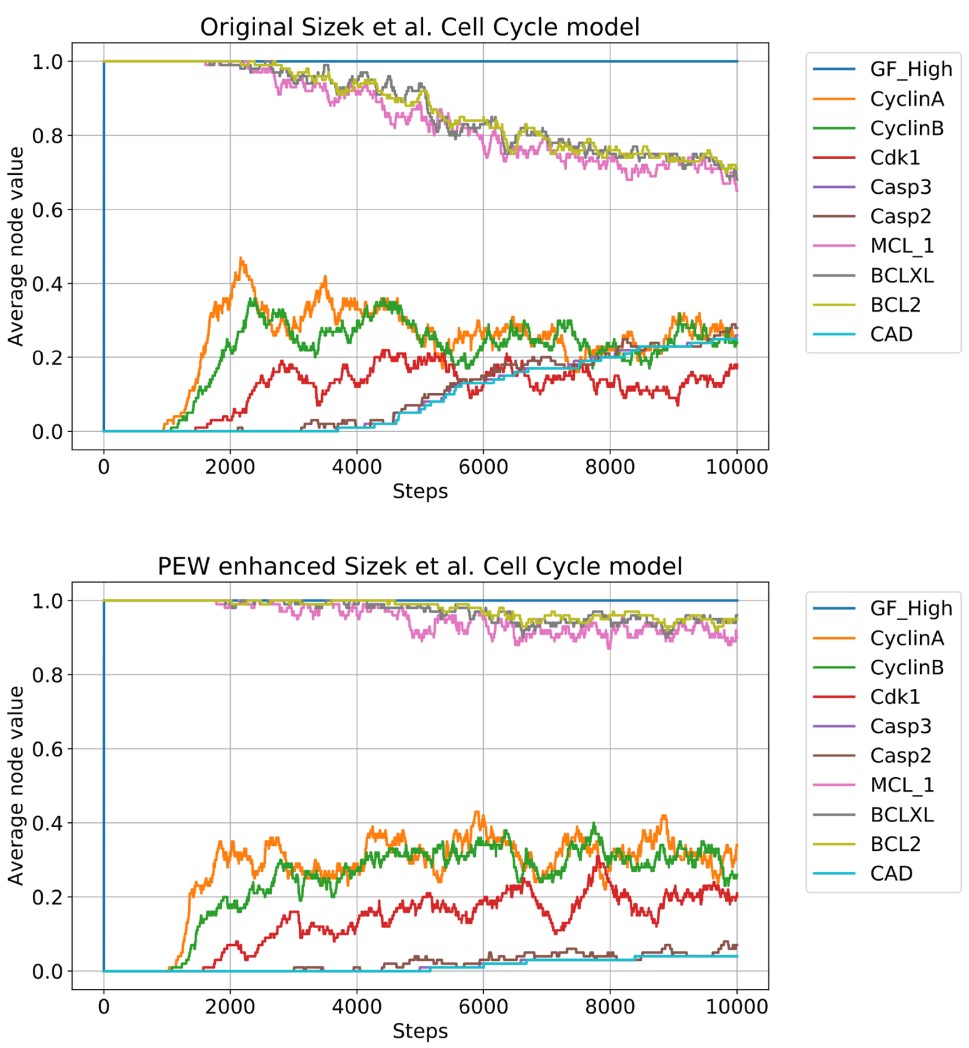

**Fig 4. The rate of cells committing to unsolicited apoptosis drops significantly after the introduction of PEW operators.** The figures show the average node values of 100 independently simulated Cell Cycle models with High growth factor stimulation and no external damage. On the top figure (original model) after 10000 steps, roughly 20% of the cells have CAD active, meaning that they committed to apoptosis. On the bottom figure (4 PEW operators added to the edges from the CyclinB-Cdk1 complex to the anti-apoptotic nodes–see S4 Document), this rate is about half of that. The same difference in apoptosis commitment rate is also visible in the evolution of anti-apoptotic nodes (MCL-1, BCLXL, BCL2). Otherwise, there is no qualitative difference in the cell cycle progression (the cell cycle markers such as CyclinA and CyclinB still fluctuate within the same boundaries, i.e. the same proportion of cells are active in the cell cycle over time).

Here we propose a simpler solution to some of the model's problems posed by the general asynchronous update scheme and a few PEW operators, while also keeping the wide range of model behaviors that match experimental data.

In Fig 4 we show an extended ensemble simulation of the original cell cycle model with constant high growth factor stimulation. This model focuses on cell cycle dynamics and assumes healthy growth conditions, i.e. no external damage or perturbation is induced. In the figure we track the concentration of only a few key nodes out of all 89. The cell cycle driver kinases and cyclins turn on early and drive the cell cycle, however, one can notice that the apoptotic nodes, such as Casp3, Casp2, and most notably CAD (which signals the execution of apoptosis) gradually increase in time—i.e. more and more cells in the ensemble die. This is

due to the several faulty behaviors related to the update scheme, which ultimately lead to apoptosis instead of continued cycling. These behaviors are discussed in Sizek et al. [30] in more detail, along with ways to overcome them using a biased update scheme.

Most often the unexpected apoptosis is the result of the perturbed balance between pro- and anti-apoptotic influences during metaphase. In metaphase cells are sensitized to apoptosis in the event of a failure to assemble the mitotic spindle and pass the spindle assembly checkpoint. There is evidence that the CyclinB-Cdk1 complex phosphorylates a part of the pool of anti-apoptotic proteins, such as MCL-1, BCLXL, and BCL2. A prolonged mitotic phase leads to a high degree of phosphorylation, which then degrades the anti-apoptotic proteins and initiates apoptosis [41–44]. This happens very rarely in healthy cells, yet with asynchronous updating, the in-silico time scales at which the event occurs are often not proportional (significantly shorter) to the time scales in real cells. To counter this we propose a weakening of the inhibitory links from the CyclinB-Cdk1 complex to the anti-apoptotic links. This modification will have no effect on the incredibly important role of the CyclinB-Cdk1 complex in driving the cell cycle, nor will it completely remove its influence on the anti-apoptotic nodes. Also, we did not alter the protective (anti-apoptotic) effects Cdk1/CyclinB has during normal cell cycle progression, where it blocks Caspase 2 activity. This means that *if* the cycle is indeed stopped during mitosis due to external damage, the CyclinB-Cdk1 complex can still prime cells for apoptosis, only it will take relatively more time (update steps). This, however, should happen very rarely in wild-type behavior (no external perturbation).

In Fig 4 (bottom) we show the effect of introducing a $[p_{on} = 0.5, p_{off} = 0]$ weight on three hyperedges from the CyclinB and Cdk1 complex to the three anti-apoptotic nodes mentioned above (MCL-1, BCLXL, and BCL2). Due to this modification, the number of cells committing to apoptosis during the same time period drops to half. In Fig 5 we present a more detailed analysis of how the ratio of apoptotic cells changes as a function of the $p_{on}$ parameter. One can

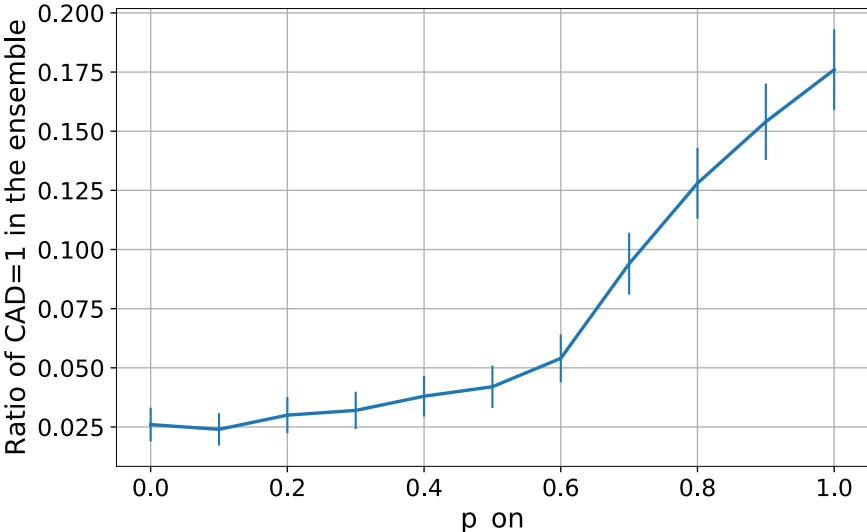

**Fig 5. The propensity of CAD = 1 (apoptosis commitment) drops with the decrease of the $p_{on}$ (increase in state-dependent noise) on the selected edges.** Each data point represents a simulation done on an ensemble of 500 models for each $p_{on}$ value with 10000 simulation steps, as a function of $p_{on}$ in the PEW operator $[p_{on}, 0]$ acting on the CyclinB +Cdk1 complex's links inhibiting the anti-apoptotic nodes (MCL-1, BCLXL, and BCL2). The same $p_{on}$ value is applied for all three hyperedges (CyclinB and Cdk1 clauses). The "default" behavior is at $p_{on} = 1$, while $p_{on} = 0$ is the special case of severing the link with the clause "Cdk1 and CyclinB "always being 0. Error bars represent the standard error within the ensemble.

notice that a 50% reduction in the CAD activity can be achieved with $p_{on} = 0.6$ and further weakening the links (which might compromise their intended function) no longer reduces the CAD activity in a significant way.

It is beyond the scope of this paper to address all the issues that are caused by the noisy update scheme in this cell cycle model. The fact that a nonzero fraction of cells still commit to apoptosis without simulated damage is due to other unintended effects of the general asynchronous update, such as aneuploidy and skipped cytokinesis. (For further details on the model's errors and explanations see Suppl. Fig 6 in [30]).

However, one can see from these simple examples, that with slight modifications of edge weights one can achieve a high degree of improvement in the quality of the emergent behavior. The method achieves this by effectively reducing the number of non-biological update orders otherwise allowed by the general asynchronous update scheme, which is especially problematic when in the real system different links work on significantly different time scales.

All the above results, for both applications, can be reproduced in the supporting S4 Notebook.

For further use-cases and applications please see the Application Note (S3 Document) and the Jupyter notebooks accompanying this manuscript (S1, S2, S3, S4 Notebooks).

## Discussion

In this paper, we have presented a method of introducing different levels of state-dependent edge-noise into Boolean models of biological systems via probabilistic weight operators. The goal of this method is not necessarily restricted to modeling the intrinsic stochasticity of systems (even though adding general noise is possible within this framework). The canonical stochastic methods discussed in the paper have a wide variety of ways of introducing system-wide as well as node-level biological noise. The goal of the PEW method is threefold.

*First*, to introduce stochasticity to the level of edges and thus introduce relative weight to edges and hyperedges through noise operators. With this, one can emulate very specific mutations, changes in binding site affinities, differences in interaction time scales, etc., and also offer a way to implement edge-level uncertainty.

*Second*, to offer a technical implementation for in-silico simulations which is easy to use; thus the method is implemented as an extended version of the well-established BooleanNet tool [33]. The *third* goal is to reproduce the noisy dynamics of several canonical methods (discussed in more detail in S2 Document). Thus, the PEW framework shouldn't be thought of as a new modeling framework, instead, as a toolkit for fine-tuning Boolean models with the targeted placement of PEW operators. Using our method one can introduce an increased uncertainty for certain connections, or conversely given a non-zero base noise, increase the relative certainty of regulations and even emulate a short-term adaptation akin to cellular learning. Cellular learning has been increasingly discussed in recent studies as a way of cells acclimatizing to repeated signals with faster phenotype convergence [45]. In this framework, we can think about learning as the reduction of noise, i.e., one could model the more optimal reaction of learning cell cultures with changing weights from noisier to less noisy parameters. Changing levels of noise within positive feedback loops could be especially effective, as we have shown in the application to the EMT study, where a less noisy edge (Zeb—miR200 in the wild type case) produced a faster epithelial convergence than the more noisy edge (mutant). In fact, there is evidence to suggest that intrinsically disordered proteins (IDPs) do exactly that, namely, acclimatize to repeated signals and shape their originally disordered (more noisy) structure to one more responsive to the signal (less noisy) [45–48].

Another reason to target self-sustaining positive feedback loops (stable motifs) with PEW operators is to counteract their intrinsic irreversibility in permanently committing the system to certain paths. For example, autocrine signaling loops involving the secretion of signaling molecules that drive their own production can create stable motifs that irreversibly lock a modeled cell into a particular self-sustaining state. Inherent in such a model is the assumption that a cell secretes a sufficiently strong signal to saturate its own autocrine signaling pathway; an assumption that may not be realistic in micro-environments that do not help concentrate these signals [22]. With PEW operators one can implement this effect without losing the non-linear effects of feedback mechanisms, as we have shown in the EMT commitment. Even having a single link made slightly noisier in a stable motif can act as a natural decay parameter as it gradually unlocks a self-sustaining feedback loop and allows different pathways to engage.

One more potential application is encoding uncertainty in data-driven regulatory network building. Many of the emerging network medicine methods generate weighted networks from combining information from different omics data sources [49,50]. Combining expression data with such regulatory networks in order to produce dynamic models is an important challenge as more and more clinical research is focused on finding therapeutic targets through the control of dynamical networks. Binarizing the weights of such data-driven networks might involve a lot of costly compromises, thus having a way of encoding edge-weights in the form of noise in a dynamic model could be a handy tool.

We would like to emphasize that this method is not necessarily restricted to the Boolean case. Indeed, a case using continuous variables was already implemented to reproduce the results of [29] (see S2 Notebook). Moreover, discrete, multi-level versions are also possible. In this case, however, one might want a different weight parameter for all possible node values if the state-dependent aspect of the method is meant to be kept.

Finally, this paper does not offer any concrete way to methodically determine the weights in the PEW operators. We believe that this is a highly context-dependent task and if the number of PEW operators introduced in a model is relatively low (like in the applications presented in this paper) the weights can be fine-tuned manually. Having too many PEW operators without a systemic source for the weights will eventually pose the same problems ODE models face, of having to fine-tune too many parameters. The collective behavior of many interacting PEW-enhanced edges should be the subject of further research.

There is an important body of work focusing on parameter estimation and control, which can be readily applied to this framework. In the case of stochastic dynamics in Boolean networks the state space is not necessarily ergodic (though it can have ergodic subsets [51]), however it can be made ergodic by adding a sort of thermodynamic noise (such as in the case of BNp) or a damping factor used in the PageRank algorithm [52], where the system can jump to random states with a nonzero probability. Murrugarra et al. [53] uses the PageRank approach to estimate the stationary probability distribution of states and then applies a genetic algorithm to estimate the node propensity parameters (akin to the PEW tuples) of the SDDS model, given a desired final probability distribution on the state space, such as more balanced attractor basins. We argue that this method can be expanded to our PEW model in a straightforward way, to estimate edge-weights.

Controlling stochastic dynamic systems is a challenging task. Aguilar et al. proposes a near optimal method in [54] for estimating control policies in stochastic Boolean networks, which similarly to the parameter estimation method by Murrugarra et al. [53], can theoretically be expanded to the PEW framework. In both the parameter estimation and the control methods the state-space of PEW Boolean models have to be made ergodic. One way of doing that is the one proposed in S2 Document where we reproduce the BNp variant with a universal noise.

Investigating the different methods of stable motif control [15,55] combined with PEW Boolean models could be a relevant next step. We show that noisy edges in stable motifs have a significant impact on the emergent dynamics, due to the fact that stable motifs are crucial in locking in attractor states.

In this paper, we introduced probabilistic edge-weights into Boolean dynamic network models and showed that they successfully model system noise making the dynamic predictions more accurate. Moreover, we developed the PEW framework, a simple-to-use implementation of this methodology in the widely used BooleanNet program package. Probabilistic edge-weight operators may open new routes to understanding the cellular learning process, as well as to include omics data to Boolean network dynamics models.

## Supporting information

**S1 Document. Discusses a comparison and conversion between PEW models and PBN (Probabilistic Boolean Networks) models.** The document discusses cases of equivalence and cases where the two approaches are different, demonstrated through examples.
(PDF)

**S2 Document. Other Noisy Boolean models as special cases of PEW.**
(DOCX)

**S3 Document.  Application Note:** The document presents three more applications, where the PEW operators offer a relatively straightforward way to encode complex edge modulation, as well as examples for the usage of the general PEW operators, where the noise function is specified as well.
(DOCX)

**S4 Document. Boolean rules of the models analyzed in the paper with and without PEW operators.**
(DOCX)

**S1 Notebook. (Murrugarra_et_al_2012_paper_results.ipynb): Jupyter notebook that reproduces some of the results of Murrugarra et al. [19].**
(HTML)

**S2 Notebook. (Poret_et_al_paper_results-Boolean_init.ipynb): Jupyter notebook that reproduces the results of Poret et al. [30].**
(HTML)

**S3 Notebook. (Supplementary_Application_Note.ipynb): Jupyter notebook that reproduces the results discussed in the Application Note (S3 Document).**
(HTML)

**S4 Notebook. (Applications.ipynb): Jupyter notebook that reproduces the results discussed in the main manuscript.**
(HTML)

## Acknowledgments

The authors thank Prof. Réka Albert and Prof. Erzsébet Ravasz Regan for their feedback and insightful comments that significantly improved the paper. The authors also thank Dr. Kimberly Glass for her support of the project.

## Code availability

The modified version BooleanNet [10] along with the Jupyter notebooks containing the code for all the results presented in this paper, as well as the ways to reproduce some of the other noisy models are available on the following GitHub page:

https://github.com/deriteidavid/boolean2pew.

(S1 Notebook: Murrugarra_et_al_2012_paper_results.ipynb, S2 Notebook: Poret_et_al_paper_results-Boolean_init.ipynb, S3 Notebook: Supplementary_Application_Note.ipynb, S4 Notebook: Applications.ipynb).

## Author Contributions

**Conceptualization:** Dávid Deritei, Nina Kunšič, Péter Csermely.

**Formal analysis:** Dávid Deritei.

**Funding acquisition:** Péter Csermely.

**Investigation:** Dávid Deritei, Nina Kunšič.

**Methodology:** Dávid Deritei.

**Software:** Dávid Deritei.

**Supervision:** Péter Csermely.

**Validation:** Dávid Deritei, Nina Kunšič.

**Visualization:** Dávid Deritei.

**Writing – original draft:** Dávid Deritei.

**Writing – review & editing:** Dávid Deritei, Nina Kunšič, Péter Csermely.

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
