## [Decision Letter · Decision Letter 0]

29 Mar 2022

Dear Dr Deritei,

Thank you very much for submitting your manuscript "Probabilistic Edge Weights Fine-tune Boolean Network Dynamics" for consideration at PLOS Computational Biology.

As with all papers reviewed by the journal, your manuscript was reviewed by members of the editorial board and by several independent reviewers. In light of the reviews (below this email), we would like to invite the resubmission of a significantly-revised version that takes into account the reviewers' comments.

We cannot make any decision about publication until we have seen the revised manuscript and your response to the reviewers' comments. Your revised manuscript is also likely to be sent to reviewers for further evaluation.

Sincerely,

Pedro Mendes, PhD

Associate Editor

PLOS Computational Biology

Mark Alber

Deputy Editor

PLOS Computational Biology

Reviewer's Responses to Questions

**Comments to the Authors:**

Reviewer #1: This manuscript presents a modeling framework that introduces noise at the edge level in Boolean networks. This framework could be useful for producing stochastic simulations of discrete models of biological systems. The authors applied their framework to two existing biological models and then showed the beneficial aspects of using their framework compared to the classic synchronous and asynchronous update modes. One limitation of this manuscript is that it does not address the problem of parameter estimation of the parameters this new framework requires. One positive aspect of the manuscript is that they provide the implementations of the discussed examples in Jupyter notebooks.

Overall, I found this manuscript mostly well-written and their results relevant. However, there are many issues in this manuscript that I suggest addressing them before recommending it for publication.

Major Revisions:

1. On page 3, in the last paragraph, it says that one of the earliest types of noisy BN were BNp in reference [27] which was published in 2013. However, PBN with noise (described in the PBN book from 2010) is much earlier than the BNp paper in reference [27]. Then, on page 4 it says PBNs address the lack of node-specificity of BNps. I think these two paragraphs were presented in a backward order. You could switch the order of the presentation by first describing PBNs as one of the earliest frameworks that introduces noise into the system and then describe that one can use BNps to make the system ergodic as in the case of PBNp.

2. The framework presented in this paper is very similar to SDDS for which there are methods for parameter estimation (see publication a) below) are well as method for control (see publication b) below). These references could be included when discussing the connections.

a) Estimating Propensity Parameters using Google PageRank and Genetic Algorithms. Frontiers in Neuroscience, 10:513, 2016

b) A Near-Optimal Control Method for Stochastic Boolean Networks. Letters in Biomathematics, 7(1), 67-80, 2020.

3. The initial states were not specified in Figure 2. Were these selected at random?

4. What does it mean that the stable motifs don’t loose their nonlinear attributes? What nonlinear attributes are you referring to in the caption of Figure 3?

5. Explain the axes of Figure 3. What is depicted in the horizontal axis?

6. Figure 4 appears on two separate pages. Merge the two panels so that they show up in the same page.

Minor Revisions:

1. On page 3, at the end of the second paragraph, it says “truly random”. It would be is best to remove the adjective “truly” and just go with random as it might create confusion.

2. It seems the acronym for the Transforming Growth Factor-beta should be TGF-beta instead of TGFB.

Reviewer #2: The author present an extension of Boolean Network Dynamics, by allowing probabilistic weights on network edges.

I am not convince that this present work is worth publishing, substantial modifications are necessary.

1) The presentation of the probabilistic grammar is clear, but the type of modeling output(s) is not described. Is it time-dependent? Probabilistic over Boolean node states? Probabilistic over the full set of Boolean node states?

2) What is the mathematical framework of this approach? Is it a discrete-time Markov chain? It seems to me that it is a special case of Probabilistic Boolean Networks, because there are different transition rules for a node, weighted by probabilities. Can the authors demonstrate that their approach is (or is not) a special case of Probabilistic Boolean Networks?

3) Although the authors show applications where their approach is efficient (in the first example) and more elegant (in the second), I am not convince that it is a major improvement of actual Boolean approaches. For that, the authors should show that a lot of biological reactions/influences are impossible to implement in any actual Boolean approaches and are only possible within their approach.

**Have the authors made all data and (if applicable) computational code underlying the findings in their manuscript fully available?**

Reviewer #1: Yes

Reviewer #2: Yes

PLOS authors have the option to publish the peer review history of their article (what does this mean?). If published, this will include your full peer review and any attached files.

Reviewer #1: No

Reviewer #2: No
---

## [Decision Letter · Decision Letter 1]

2 Sep 2022

Dear Dr Deritei,

We are pleased to inform you that your manuscript 'Probabilistic Edge Weights Fine-tune Boolean Network Dynamics' has been provisionally accepted for publication in PLOS Computational Biology.

Best regards,

Pedro Mendes, PhD

Academic Editor

PLOS Computational Biology

Mark Alber

Section Editor

PLOS Computational Biology

Reviewer's Responses to Questions

**Comments to the Authors:**

Reviewer #1: The authors have addressed all my comments from the previous round of reviews. There is a small typo in the Author Summary: change "forunderstanding" to "for understanding". This can be implemented during the copy editing stage.

Reviewer #2: The author answer to all my comments in an efficient way. They modify the text accordingly; perhaps they should more explicitly describe the type of output they use: their ensemble average.

Nevertheless, I am not convince that this approach consists of an important enhancement of Boolean modeling. According to the authors, it is a generalization of PBN, with a more flexible choice of the noise function. In particular, this allows to consider partial inhibition of edges, in a more efficient way than for PBN. To my opinion, this has not a wide applicability. Why not just implement their approach in PBN existing tools?

**Have the authors made all data and (if applicable) computational code underlying the findings in their manuscript fully available?**

Reviewer #1: Yes

Reviewer #2: Yes

PLOS authors have the option to publish the peer review history of their article (what does this mean?). If published, this will include your full peer review and any attached files.

Reviewer #1: No

Reviewer #2: No

---

## [Editor Report · Acceptance letter]

5 Oct 2022

PCOMPBIOL-D-22-00184R1 

Probabilistic Edge Weights Fine-tune Boolean Network Dynamics

Dear Dr Deritei,

I am pleased to inform you that your manuscript has been formally accepted for publication in PLOS Computational Biology. Your manuscript is now with our production department and you will be notified of the publication date in due course.

With kind regards,

Zsofi Zombor
